# Targeting P53 as a Future Strategy to Overcome Gemcitabine Resistance in Biliary Tract Cancers

**DOI:** 10.3390/biom10111474

**Published:** 2020-10-23

**Authors:** Chiao-En Wu, Yi-Ru Pan, Chun-Nan Yeh, John Lunec

**Affiliations:** 1Division of Hematology-Oncology, Department of Internal Medicine, Chang Gung Memorial Hospital at Linkou, Chang Gung University College of Medicine, Taoyuan 333, Taiwan; jiaoen@gmail.com; 2Department of General Surgery and Liver Research Center, Chang Gung Memorial Hospital, Linkou branch, Chang Gung University, Taoyuan 333, Taiwan; panyiru0331@gmail.com; 3Newcastle University Cancer Centre, Bioscience Institute, Medical Faculty, Newcastle University, Newcastle upon Tyne NE2 4HH, UK

**Keywords:** p53, gemcitabine resistance, biliary tract cancer

## Abstract

Gemcitabine-based chemotherapy is the current standard treatment for biliary tract cancers (BTCs) and resistance to gemcitabine remains the clinical challenge. *TP53* mutation has been shown to be associated with poor clinicopathologic characteristics and survival in patients with BTCs, indicating that p53 plays an important role in the treatment of these cancers. Herein, we comprehensively reviewed previous BTC preclinical research and early clinical trials in terms of p53, as well as novel p53-targeted treatment, alone or in combination with either chemotherapy or other targeted therapies in BTCs. Preclinical studies have demonstrated that p53 mutations in BTCs are associated with enhanced gemcitabine resistance, therefore targeting p53 may be a novel therapeutic strategy for treatment of BTCs. Directly targeting mutant p53 by p53 activators, or indirectly by targeting cell cycle checkpoint proteins (Chk1, ataxia telangiectasia related (ATR), and Wee1) leading to synthetic lethality, may be potential future strategies for gemcitabine-resistant p53 mutated BTCs. In contrast, for wild-type p53 BTCs, activation of p53 by inhibition of its negative regulators (MDM2 and wild-type p53-induced phosphatase 1 (WIP1)) may be alternative options. Combination therapies consisting of standard cytotoxic drugs and novel small molecules targeting p53 and related signaling pathways may be the future key standard approach to beat cancer.

## 1. Current Treatment for Biliary Tract Cancers (BTCs): Chemotherapy, Targeted Therapy, and Immunotherapy

Biliary tract cancers (BTCs) including intrahepatic cholangiocarcinoma (iCCA), extrahepatic cholangiocarcinoma (eCCA), gallbladder cancer, and ampullary cancer are a group of relatively rare cancers arising from the epithelium of the biliary tract [1,2,3,4] and have aggressive biological behavior, as they are diagnosed at an advanced stage with poor prognosis and have a high recurrence rate after primary surgery [5,6]. Gemcitabine and cisplatin have been the standard treatments in first-line chemotherapy since the ABC-02 trial was published in 2010 [7]. Clinical trials have evaluated molecular targeted therapies in combination with chemotherapy; however, none of the completed phase III trials [8,9,10,11] and phase II studies [12,13,14,15] have demonstrated significant improvement in progression-free survival (PFS) and overall survival (OS) in patients with advanced BTCs [16]. Real word experience has confirmed the feasibility and safety of gemcitabine and cisplatin in advanced BTCs [6,17].

With the development of comprehensive genetic profiling by next generation sequencing (NGS), alterations of isocitrate dehydrogenase genes (IDH1, IDH2) and fibroblast growth factor receptors (FGFR1, FGFR2, FGFR3) were found in iCCA and targeting such genetic alterations has become a novel therapeutic strategy in iCCA [18,19]. Recently, pemigatinib (Pemazyre), a FGFR inhibitor, has demonstrated activity in previously treated, unresectable locally advanced or metastatic CCA with FGFR2 fusions or rearrangements [20]. Such FGFR2 alternations account for 15% of iCCA [19] and are not found in other BTCs, limiting its general application in BTCs. In addition, pemigatinib is approved to be used for chemotherapy-treated CCA patients, so chemotherapy is still the first-line standard treatment for iCCA patients.

Although immune checkpoint inhibitors (ICIs) showed limited efficacy in unselected patents [21], a number of biomarker-guided immunotherapy studies are currently recruiting for ongoing trials [22]. Pembrolizumab was approved for treatment of a variety of advanced microsatellite instability-high (MSI-H) or deficient mismatch repair (dMMR) solid tumors (including BTC) [23] in 2017, but MSI-H or dMMR rarely occur in BTCs, including iCCA [24,25]. Recently, pembrolizumab was approved for treatment of tumor mutation burden-high (TMB-H) solid tumors, which is defined as ≥ 10 mutations/megabase (mut/Mb) assessed by FoundationOneCDx assay. Similar to MSI-H/dMMR, TMB-H rarely happens in BTCs, limiting the utility of ICIs in BTCs. Combination trials of ICIs and chemotherapy are ongoing to explore the potential for future application in BTCs.

Therefore, chemotherapy is still the mainstay of treatment in advanced BTCs. In CCA patients with FGFR2 fusion or rearrangement, pemigatinib is a late-line treatment option. The use of ICIs such as pembrolizumab is limited to CCA patients with MSI-H/dMMR or TMB-H.

## 2. Introduction to p53 in Cancer

In 1979, p53 was originally discovered by several groups as the host tumor protein targeted by the large tumor antigen of SV40 DNA tumor virus [26,27,28,29,30]. The *TP53* gene, which encodes p53, was generally considered to be an “oncogene” before the mid-1980s [31,32,33], when sequencing of p53 cDNA clones revealed that some of them carried mutations that were comparable to the sequence of murine wild-type p53 derived from normal tissues [34,35]. It became clear that p53 mutations are frequently present in tumor-derived cell lines [34] and it was found that mouse tumor derived mutant (MUT) p53 could indeed promote cell transformation and survival but wild-type (WT) p53 did not. Deletion mapping studies accompanied by DNA sequencing demonstrated a frequent two-hit p53 allele inactivation mechanism in human tumors and strongly indicated that p53 works as a tumor suppressor [36]. The observation that transformation of cultured cells could be repressed by overexpression of WT p53 confirmed its tumor suppressor role [37]. Further confirmation of the tumor suppressor role of p53 and the two-hit inactivation mechanism came from the demonstration of mono-allelic germline *TP53* mutations in Li-Fraumeni familial cancer predisposition syndrome, and the second hit inactivation was evident in the tumors of the affected Li-Fraumeni patients [38].

In the early 1990s, p53 was found to be a transcription factor [39] that can bind tightly to a specific DNA consensus sequence [40]. The abilities to bind to specific sequences and transactivate particular genes distinguishes WT p53 from all cancer-associated p53 mutants. *CDKN1A* encoding the cyclin-dependent kinase inhibitor p21 [41] and the proapoptotic *BAX* gene [42] were found to be transactivated by p53. Subsequently, a number of additional p53 transcriptionally targeted genes have been identified, most of which encode proteins that are intimately involved in apoptosis or in control of cell cycle progression. In addition, two of the transcriptionally targeted genes, *MDM2* and *PPM1D*, have an autoregulatory feedback role to regulate p53.

The MDM2 protein, the most important protein directly interacting with p53, was discovered in 1992. MDM2, an E3 ubiquitin ligase, was shown to bind to p53 tightly and inhibit p53 transcriptional activity [43] as well as promoting the ubiquitylation and subsequent proteasomal degradation of p53 [44]. MDMX (MDM4 in the mouse) was discovered as a paralogue of MDM2 in 1996, which can form heterodimers with MDM2 to augment MDM2 activity and contribute to p53 degradation but has no measurable intrinsic E3 ligase activity [45,46]. The intimate relationship between MDM2 and p53 was demonstrated by gene knockout studies in mice. Knockout of the *MDM2* gene was found to be embryonically lethal but could be rescued by additional knockout of the mouse p53 gene [47].

Activation of p53 occurs in response to cellular stress, particularly as a consequence of DNA damage response (DDR), such as induced by gemcitabine. The DDR signaling results in phosphorylation of p53 and MDM2, which prevents MDM2 from binding to p53. This allows levels of the p53 protein to increase and activates its function as a transcription factor [48,49] to drive the expression of p53 target genes, which execute the appropriate cellular responses, including DNA repair, altered metabolism, cell cycle arrest, or apoptosis and senescence [50]. The *TP53* gene is one of the most frequently mutated genes in human cancer [51] and has been found to be either mutated or the p53 protein functionally inactivated in most human cancers [52,53]. Homozygous mutation and/or deletion of *TP53* results in loss of WT p53 tumor suppressor function. However, some of the point missense mutations have also been shown to have dominant oncogenic functions (gain-of-function mutations) that can override WT p53 from a remaining WT *TP53* allele, via binding to other transcription factors that transactivate genes associated with tumor survival and drug resistance [54] (Figure 1).

## 3. p53 in BTCs

### 3.1. p53 Mutation in BTCs

In most cancers, *TP53* is one of the top frequently mutated genes; however, iCCA have p53 mutations only in a minority of cases (23%) [18], indicating that 77% of iCCA have WT p53, which nevertheless may be repressed by its negative regulators. In contrast, the frequency of *TP53* mutations in subtypes of BTC, eCCA and GB has been reported to have a higher frequency of *TP53* mutation than iCCA (Figure 2B). Overall, *TP53* mutations are present in around 32% of all BTCs (Figure 2C). *TP53* mutations frequently occur in the DNA binding domain of p53, resulting in loss of transcriptional transactivation function (Figure 2D).

In addition to mutation, the function of p53 can be suppressed by *MDM2* amplification, or induced MDM2 activity due to loss of its negative regulator, p14ARF, as a result of deletion of the *CKDN2A* locus. Therefore, not only p53 but also the integrity of the CKDN2A(p14ARF)/MDM2/p53 axis may play an important role in BTC (Figure 3). Alterations of MDM2 (mostly amplification) and CKDN2A (mostly deep deletion) occur in 4% and 10% of BTCs, respectively (Figure 3A). Furthermore, *TP53* mutation and MDM2 alteration are mutually exclusive (*p* = 0.011) in BTCs, implying either gene alteration may be sufficient to abrogate p53 function leading to tumorigenesis (Figure 3B).

### 3.2. p53 in Tumorigenesis of BTCs

Unlike other cancers, the specific role of *TP53* mutation in BTC tumorigenesis has not yet been well established, possibly due to the relative rarity compared with other cancers. Importantly, liver fluke- and/or viral hepatitis-positive BTCs have a higher incidence of *TP53* mutations, reflecting distinct genetic alterations associated with different aetiologies [55,56,57]. Therefore, to better understand the tumorigenesis of BTC, comprehensive genome sequencing is needed for BTC patients from different environments associated with different pathological factors.

### 3.3. p53 as a Prognostic Factor in BTCs

Early studies investigated the prognostic role of p53 protein expression by immunohistochemistry (IHC) and/or *TP53* mutation in BTCs and the results are conflicting [58]. Some studies showed no association with survival and some showed p53 protein expression or *TP53* mutation negatively impact survival [58]. However, one study showed p53 protein expression was associated with better survival [59]. It should be cautioned that the relationship between protein detection by IHC and p53 mutational status is complex. Some p53 mutant proteins resulting from point missense mutations are more stable than p53^WT^ when they result in conformational alteration, which leads to less transactivated MDM2 and also less affinity with MDM2. Such p53 mutants can be detected by strong IHC staining for p53. However, the p53 mutants resulting from nonsense mutations or frameshift insertions/deletions may not be detected if the p53 was simply examined by IHC. Therefore, gene sequencing is a more reliable method to indicate the potential loss of normal p53 function and, in some cases, dominant abnormal gain of function.

With the advent of next generation sequencing (NGS), comprehensive genetic profiling of BTCs has been investigated by different study groups. Based on the pooled data, patients with BTCs harboring *TP53* mutation have a poorer survival outcome than patients with *TP53* WT BTCs (Figure 4A,B). In addition, the combination of *CDKN2A/MDM2/TP53* alterations indicating loss of p53 function provide a more comprehensive prediction of poor prognosis in patients with BTCs (Figure 4C,D). Due to the heterogeneity of patient characteristics and treatment in collected studies, sufficiently powered studies with more uniformly defined cohorts of patients are needed in the future to validate this finding.

### 3.4. MDM2 as a Prognostic Factor in BTCs

MDM2 is a negative regulator of p53 and is also transcriptionally transactivated and tightly regulated by p53. Therefore, MDM2 is expected to be overexpressed only in p53^WT^ tumors and MDM2 overexpression may be an alternative mechanism to suppress the function of p53^WT^, potentially resulting in aggressive tumor behaviors. An early report of 47 iCCA patients showed that MDM2 overexpression correlated with the Ki-67 labeling index (*p* < 0.03), presence of metastases (*p* < 0.01) and advanced tumor stage (*p* < 0.05) [60]. Another study enrolled 128 CCA patients in Thailand and evaluated the clinical significance of CD44 and MDM2. This study demonstrated that expression of CD44 and MDM2 in CCA correlated with poor clinicopathologic characteristics (high grade, large size, regional and distant metastases) and poor survival [61]. CD44 expression was associated with *TP53* mutation and tumor stemness [62] and MDM2 expression could be taken to indicate tumors harboring p53^WT^, which is nevertheless being suppressed by MDM2. Therefore, both biomarkers may possibly reflect the nature of loss-of-function of p53^WT^, which is compatible with previous studies about the significance of *TP53* mutation status in BTCs. However, this study did not analyze *TP53* mutation, so that the real relationship between CD44/MDM2/TP53 was unknown, particularly for the tumors with CD44/MDM2 co-expression.

## 4. p53 Plays a Role in Gemcitabine Resistance in Cancers

Gemcitabine treatment induces a cellular DNA damage response (DDR) signaling cascade involving activation by phosphorylation of ataxia telangiectasia mutated (ATM), ataxia telangiectasia related (ATR) and p53. Once p53^WT^ is activated by phosphorylation and subsequent accumulation, it induces cell cycle arrest, allowing and facilitating DNA repair, or, depending on the cell type and level of damage, it triggers cell death (by apoptosis or alternative mechanisms) or cell senescence. However, in p53^MUT^ cancer, p53 loses its WT function and in some cases may gain MUT function, either way this allows survival rather than growth inhibition, but the lack of cell cycle arrest also results in replication of DNA on a damaged template and hence further mutations. In some cases of p53^WT^ cancer, WT p53 is suppressed by its negative regulators such as MDM2 and wild-type p53-induced phosphatase 1 (WIP1) (PPM1D). Either p53^MUT^ or p53^WT^ suppression by MDM2 is proposed to increase gemcitabine resistance (Figure 5) [63].

However, studies regarding the role of p53 in gemcitabine resistance in BTCs are limited. As both pancreatic cancer and BTC have close anatomic location and similar histologic features, both were considered together in early studies. Palliative chemotherapy was demonstrated to improve both survival and quality of life for patients with advanced pancreatic cancer or BTC [64]. Gemcitabine became the standard treatment for both pancreatic cancer [65] and BTC [7]; therefore, studies related to gemcitabine resistance in pancreatic cancer may be applicable to BTCs and hence are discussed here.

Fiorini et al. demonstrated that p53^MUT^ enhanced gemcitabine resistance in pancreatic cancer. Gemcitabine stabilized both nuclear p53^WT^ and p53^MUT^ and only the latter (p53^MUT^) induced chemoresistance and hyperproliferation evidenced by siRNA-mediated knockdown induced chemosensitivity and apoptosis in p53^MUT^ but not in p53^WT^ cell lines. This may be associated with overexpression of p53^MUT^-dependent *Cdk1* and *CCNB1* genes. However, a synergistic effect of the combination with gemcitabine and “p53-reactivating” compounds (CP-31398 and RITA) was found in both p53^WT^ and p53^MUT^ pancreatic cancer cell lines, which may have been due to a mechanism that did not involve p53 [66]. In a mouse model of pancreatic cancer, Wornann et al. reported that loss of p53^WT^ function increased resistance to gemcitabine via activation of the JAK2–STAT3 pathway [67]. In a study by Dhyat et al. (2016) microRNA (miR) profiling in gemcitabine-resistant p53^MUT^ pancreatic cancer cell lines suggested the involvement of miRs in pathways controlling cell cycle, proliferation, and apoptosis. In-silico analysis in this study indicated that some of the dysregulated miRs were regulated by p53^MUT^. In addition, MRP-1, Bcl-2 and CDK-1 were predicted to be targets of the dysregulated miRs and significant overexpression of MRP-1 and Bcl-2 was seen in the resistant cell clones [68]. These observations suggest that p53^MUT^ may enhance chemoresistance via the dysregulation of miRs and that this might be associated with MRP-1 and Bcl-2 overexpression in p53^MUT^ pancreatic cancer.

In the clinical setting, a phase III study (CONKO-001) evaluated the efficacy of adjuvant gemcitabine in resected pancreatic cancer. Next generation DNA sequencing was performed in this trial to identify clinically relevant prognostic and predictive mutations. In untreated patients, *TP53* mutation was a negative prognostic factor for disease-free survival (DFS) (HR: 2.434, *p* = 0.005). Interestingly, however, *TP53* mutation was a positive predictive factor for better DFS (HR: 0.235; *p* < 0.001 in p53^MUT^; HR: 0.794, *p* = 0.483 in p53^WT^) with gemcitabine treatment, with a significant test for interaction of the treatment status with TP53 mutational status (*p* = 0.003) [69]. Only subsets of patients with available samples were retrospectively analyzed, limiting the significance of this study, but the findings provide evidence for the role of p53^MUT^ in pancreatic cancer, which may be relevant to BTC.

Further studies specific to BTC regarding p53 and chemoresistance are warranted to provide additional evidence, which would guide treatment. However, the above studies suggest a novel therapeutic strategy targeting both wild-type and mutant p53 tumors in combination with gemcitabine for pancreatic cancer as well as BTCs.

## 5. Targeting p53 as a Therapeutic Strategy

To overcome the p53-associated chemoresistance, p53^WT^ and p53^MUT^ situations should be considered individually as they may involve different chemoresistance mechanisms (Figure 6).

### 5.1. Targeting Mutant p53

#### 5.1.1. Direct Targeting of Mutant p53

Unlike most targeted therapy against oncoproteins resulting from oncogene alterations, targeting mutant p53 has been a challenge because p53 behaves as a tumor suppressor protein, where loss of function is the predominant mechanism, but also a subset of mutations result in dominant negative oncogenic behavior. Therefore, ideally not only inhibiting p53^MUT^ gain of function mutation but also activating p53^WT^ function is necessary for BTCs. If only p53^MUT^ is suppressed by targeted therapy, the p53^WT^ may also need to be activated to maximize the efficacy of p53-targeted therapy. In addition, although the p53 mutations are frequently found in exons four to nine, which encode the DNA-binding domain of p53, no unique hotspots are available for targeting in all cases. APR-246, a PRIMA-1 analog, has been developed to covalently bind to cysteines of the core domain and modify the core domain of p53^MUT^ to restore both the WT conformation and function via conversion to the reactive electrophile methylene quinuclidinone, in order to reconstitute endogenous p53^WT^ activity, in principle leading to cell cycle arrest and apoptosis of p53^MUT^ tumor cells (Figure 6) [70]. Although the mechanism and specificity of APR-246 for p53^MUT^ tumor cells has been questioned, the FDA granted breakthrough therapy designation for APR-246 in combination with azacitidine for the treatment of patients with myelodysplastic syndromes (MDS) with a susceptible *TP53* mutation based on a promising early phase II study (NCT03588078)[71]. A phase III study is ongoing to compare the rate of complete remission and duration of complete remission in patients with p53^MUT^ MDS receiving APR-246 and azacitidine or azacitidine alone (NCT03745716). As the results in MDS have been taken to support the concept of reactivating susceptible forms of p53^MUT^, clinical trials regarding APR-246 alone or in combination with chemotherapy are ongoing in a range of hematologic and solid malignancies.

Preclinical and clinical studies of p53 activators including APR-246 are being widely investigated. APR-246 has been reported to overcome chemoresistance to cisplatin and doxorubicin and to synergize with gemcitabine in p53^MUT^ ovarian cancer cells [72]. Furthermore, combination of APR-246 with dexamethasone or doxorubicin was reported to show synergistic effects in multiple myeloma cell lines and primary multiple myeloma samples [73]. The efficacy and safety of such strategies targeting p53^MUT^ should be confirmed in clinical trials and may be applicable for BTC treatment in the future (NCT04383938).

#### 5.1.2. Indirect Targeting of Mutant p53 by Synthetic Lethality

Another possible therapeutic option is to target cell cycle regulators in p53^MUT^ cancer as a potential synthetic lethality strategy to kill p53^MUT^ cancer cells selectively. It has been hypothesized that cancer cells deficient in p53 function, due to p53 mutations or other defects in the p53 signaling pathway, lack a G1 cell cycle checkpoint and therefore may depend more on the G2 checkpoint for cell survival [74,75,76]. Abrogation of the G2 checkpoint by inhibitors of Chk1 [74], Wee1 [77], or ATR [78] may therefore selectively sensitize p53 defective cancer cells to DNA damaging agents resulting in cell death by mitotic catastrophe [79], while relatively sparing the surrounding normal p53-proficient cells. In addition, Chk1, ATR, and Wee1 inhibitor treatment has been reported to result in DNA damage in S phase, which may contribute to the cytotoxic effects of these inhibitors (Figure 6) [80,81]. Taking in vitro studies of lung cancer for example, the Chk1 inhibitor UCN-01 [82], Wee1 inhibitor AZD1775 [83], ATR inhibitor VX-970 [84] in combination with cytotoxic agents (irradiation or cisplatin) have been shown to selectively target p53-deficient lung cancer cells, demonstrated using otherwise isogenic p53^MUT^ and p53^WT^ paired cell systems.

Adavosertib (AZD1775), a Wee1 inhibitor, is currently being investigated in clinical trials. In preclinical studies, AZD1775 was reported to selectively radiosensitize p53^MUT^ but not in p53^WT^ cell lines [85]. In contrast, another study demonstrated AZD1775-mediated potentiation of antimetabolite chemotherapeutics independent of *TP53* status in both hematologic and solid tumor models [86]. Therefore, p53 dependence may be restricted to certain cancers and/or treatment modalities. Biomarker analysis in a phase one study of AZD1775 showed response rates of 21% and 12% for *TP53*^MUT^ and *TP53*^WT^ tumors, respectively, indicating a degree of selection for *TP53*^MUT^ cancers [87]. Phase II studies of AZD1775 alone and in combination with other cytotoxic agents are ongoing and larger cohort studies are needed to confirm whether p53 mutation is a predictive factor for use of AZD1775. Several clinical trials of AZD1775 are ongoing in various cancers (NCT01748825, NCT04462952, NCT02482311).

AZD6738, an ATR inhibitor, has shown preclinical activity in panels of BTC cell lines, particularly with SNU478 and SNU869 cell lines (both p53^MUT^) [88]. The combination of AZD6738 and MK1775 has also been reported to have synergistic effects in p53^MUT^ BTC cell lines [89]. Early phase studies of AZD6738 are ongoing in various cancers [90] and also AZD6738 in combination with durvalumab in BTCs (NCT04298008).

Therefore, targeting cell cycle checkpoints to induce mitotic catastrophe cell death in p53^MUT^ cancers is an emerging strategy that is being explored in the clinical setting.

### 5.2. Targeting Wild-Type p53

In p53^WT^ cancers, the p53 function is held back by its negative regulators. Therefore, inhibition of negative regulators to activate p53 function should be investigated in p53^WT^ cancers.

#### 5.2.1. Targeting MDM2, an E3 Ligase and Negative Regulator of p53

MDM2, a transcriptional target of p53, is the most important negative regulator of p53. It binds to p53 and blocks its transcriptional activity, as well as targeting p53 for degradation by ubiquitination, thus inhibiting and controlling the growth inhibitory and pro-apoptotic activity of p53. Hence, MDM2 can limit the effectiveness of p53 and p53-dependent therapies, and thus promote the development and survival of cancer cells [50]. The essential role of MDM2 in the negative regulation of p53 was highlighted in early studies, which showed that germline gene knockout of *MDM2* in mice was embryonically lethal, but could be rescued by co-deletion of the mouse gene encoding p53 [47]. The MDM2-p53 binding antagonists were developed to bind to MDM2 in the p53-binding pocket, allow p53 to accumulate, and then activate the p53 pathway in cancer cells, resulting in cell cycle arrest and apoptosis or senescence, depending on the cell type. This was first demonstrated with Nutlin-3, which was shown to inhibit proliferation and induce cell death for human cancer cell lines in vitro and for tumor xenografts in nude mice [91]. Subsequently, a number of MDM2 antagonists (RG7388 (Idasanutlin), HDM201, AMG232, etc.) have been developed and studied preclinically in vitro and in vivo. A number of these agents have gone on to be investigated in clinical trials (NCT02143635, NCT03362723).

Current investigations of MDM2 inhibitors are exploring their use in combination treatments with cytotoxic agents or targeted therapy [63]. The combination of MDM2 inhibitors and gemcitabine has been studied in p53^WT^ mantle cell lymphoma using MI-63 and MI-219 in vitro and in vivo. This combination treatment also demonstrated promising results for primary patient samples ex vivo [92].

#### 5.2.2. Targeting MDM2/MDMX

MDMX is a paralogue of MDM2 and has been reported to augment MDM2 activity and contribute to p53 degradation [93]. Compared with MDM2/p53 antagonists, the agents designed to target MDMX/p53 are relatively limited and, despite extensive efforts, small molecule inhibitors specific for MDMX have yet to be developed, although there are some compounds that inhibit both MDM2 and MDMX. For example, RO5963 has been demonstrated to inhibit p53 binding to MDM2/MDMX, and MDMX overexpression was associated with an apoptotic response to RO5963 treatment [94]. Preclinical studies have reported the activity of dual MDM2/MDMX inhibitors [95,96] and currently the ALRN-6924 peptide, alone or in combination with cytotoxic agents, is being investigated in early phase clinical trials (NCT03654716, NCT03725436).

#### 5.2.3. Targeting WIP1, a Phosphatase of p53

The *PPM1D* gene that encodes wild-type p53-induced phosphatase 1 (WIP1) is another important homeostatic negative regulator of p53 function and stability, by both directly and indirectly controlling the phosphorylation status of p53 after cellular stress. The *PPM1D* gene is also a direct transcriptional target of p53, thus forming a negative auto-regulatory loop with the p53 network by dephosphorylating p53 (Ser15) and other DDR signaling components (such as ATM, ATR, and MDM2) involved in p53 post-translational regulation [97].

GSK2830371 is a small molecule allosteric inhibitor, which has been selected to block the enzymatic activity of WIP1. It binds to a flap subdomain of WIP1 and also enhances ubiquitin-mediated degradation of WIP1 [98]. Pre-clinical studies have shown that GSK2830371 can enhance p53-mediated tumor suppression by MDM2 inhibitors [99,100] or by chemotherapy [101]. Although GSK2830371 is a useful tool compound for exploratory proof of concept studies, this compound does not have in vivo properties suitable for clinical development and the lack of suitable WIP1 inhibitors currently precludes investigation of this target in clinical trials [102].

## 6. Conclusions

*TP53* mutation is associated with adverse clinicopathological characteristics and poor survival of BTC patients. Preclinical studies have demonstrated that p53 mutation enhances gemcitabine resistance; therefore, targeting mutant p53 may be a novel therapeutic strategy for treatment of BTCs. Directly targeting mutant p53 by p53 activators or indirectly targeting cell cycle checkpoints to exploit potential synthetic lethality are strategies to explore for gemcitabine-resistant p53^MUT^ BTCs. In contrast, for the majority of *TP53*^WT^ tumor cases, activation of p53 by inhibition of its negative regulators (MDM2 and WIP1) is an option worthy of investigation.

In the future, combination therapies consisting of cytotoxic drugs with established activity in BTC and novel small molecules targeting p53 may be the key to more effective treatment of these cancers (Table 1). Overcoming resistance to traditional anticancer drugs, including gemcitabine, by development of synergistic combinations with novel targeted small molecules has the potential to improve the activities and outcomes of chemotherapy in BTC.

## Figures and Tables

**Figure 1 biomolecules-10-01474-f001:**
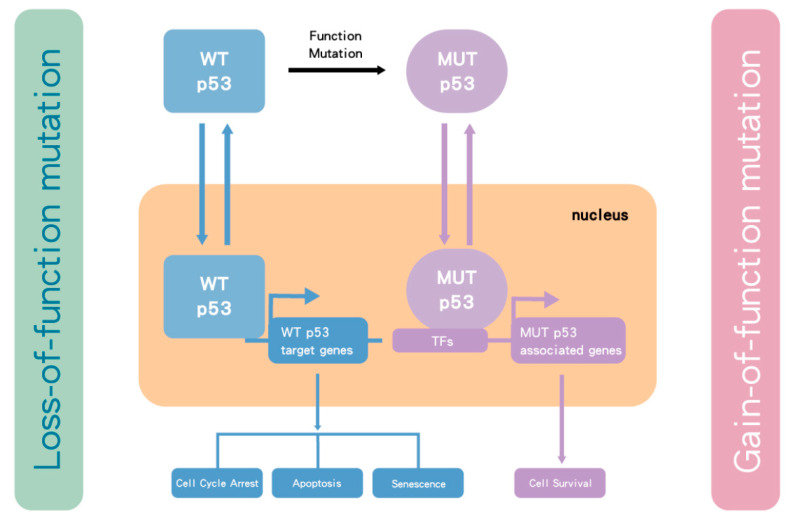
**The influence of *TP53* mutation in cancer cells.** p53 is a transcription factor that transactivates downstream genes responsible for cell cycle arrest, apoptosis and cell senescence. Once p53 becomes mutated (MUT), it loses wild-type (WT) p53 function. In some cases, the mutant form selectively accumulates due to loss of binding to MDM2. As p53 functions as a tetramer, the mutant and functionally defective form then dominates over the lower number of wild-type p53 protein molecules from the remaining normal allele. However, for the most part, *TP53* behaves as a classical tumor suppressor gene and both alleles need to be inactivated by a combination of mutation and/or deletion. Nevertheless, some mutant forms of p53 bind to transcription factors (TFs), which transactivate genes responsible for tumor survival and drug resistance. Taken together, MUT p53 enhances tumor growth.

**Figure 2 biomolecules-10-01474-f002:**
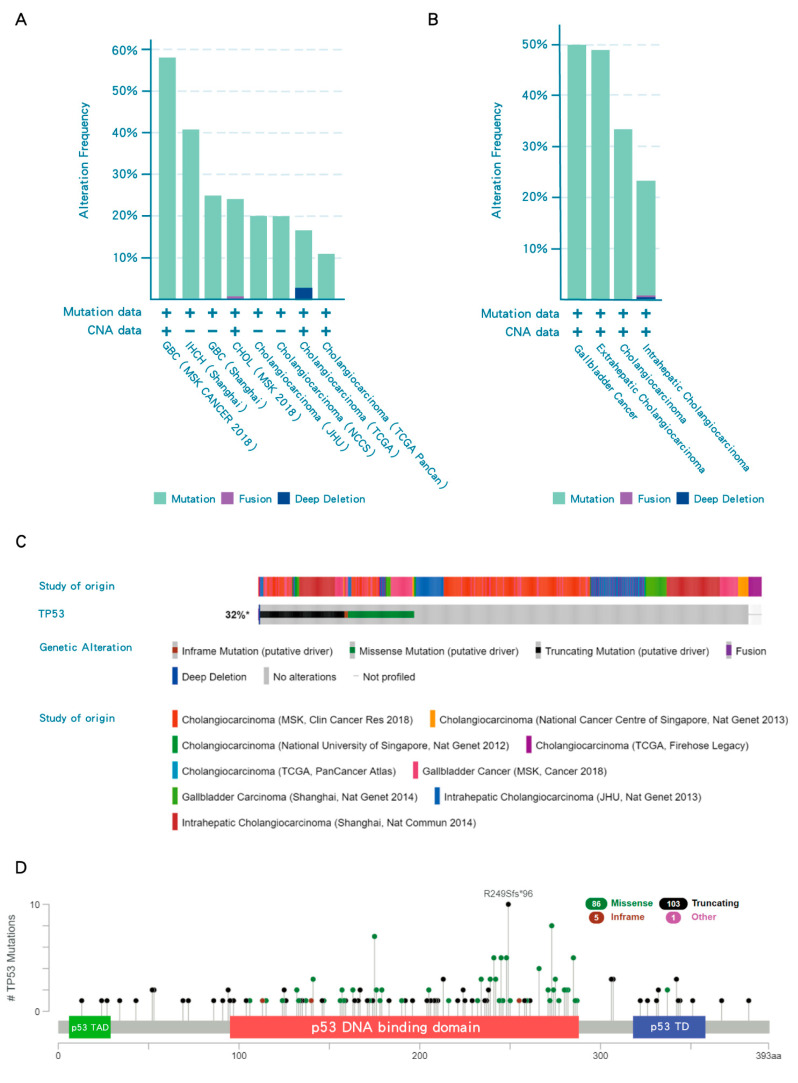
***TP53* mutations in biliary tract cancers (BTCs).** (**A**) Frequency of *TP53* mutations in different studies. (**B**) The frequency of *TP53* mutations in subtypes of BTC. (**C**) Collective results of available datasets of *TP53* mutation in BTCs. (**D**) Lollipop plots for tumor *TP53* mutations in BTC patients. *TP53* mutations frequently occur in the DNA binding domain of p53 (red region). Black, green, pink and blue dots indicate truncating, missense, in-frame and other mutations, respectively. When different mutation types occur at a single position, the color of the circle indicates the most frequent mutation type. TAD, transactivation domain; TD, tetramerization domain. Data were accessed at cBioPortal on 3 September 2020. Please add explanation for “+, −” in image.

**Figure 3 biomolecules-10-01474-f003:**
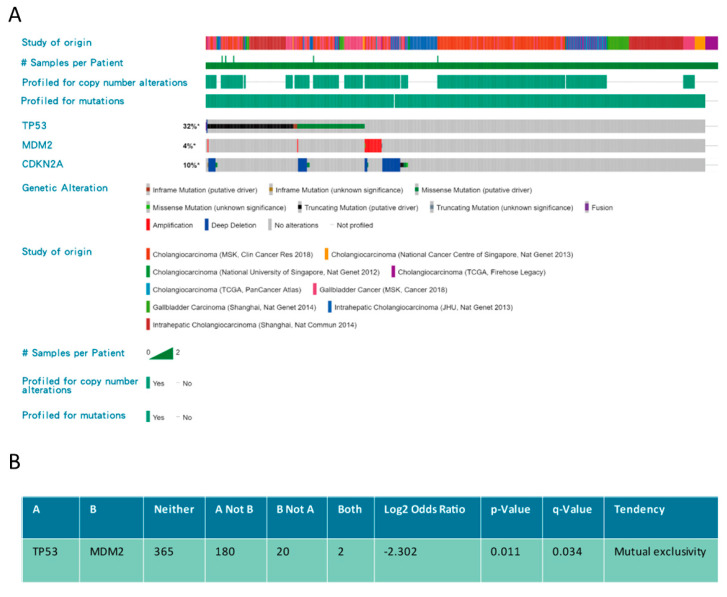
***TP53/MDM2/CDKN2A* alterations in biliary tract cancers (BTCs).** (**A**) Frequency of *TP53/MDM2/CDKN2A* alterations in different studies. (**B**) The association between *TP53/MDM2/CDKN2A* alterations. Data were accessed at cBioPortal on 3 September 2020.

**Figure 4 biomolecules-10-01474-f004:**
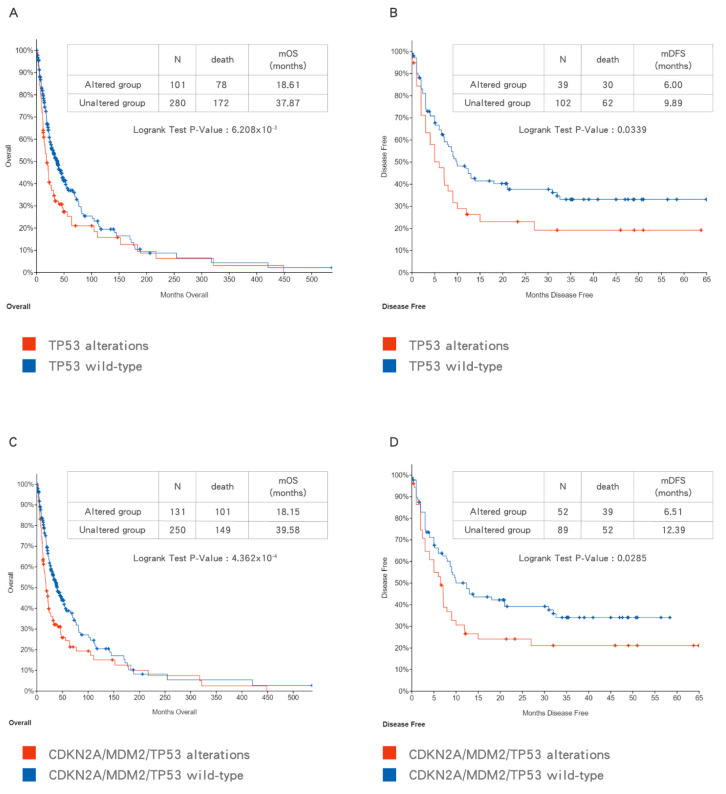
The impact of *TP53* mutation (**A**,**B**) and *CDKN2A/MDM2/TP53* alterations (**C**,**D**) on overall survival (OS) (**A**,**C**) and disease-free survival (DFS) (**B**,**D**) in biliary tract cancers (BTCs). Data were accessed at cBioPortal on 3 September 2020.

**Figure 5 biomolecules-10-01474-f005:**
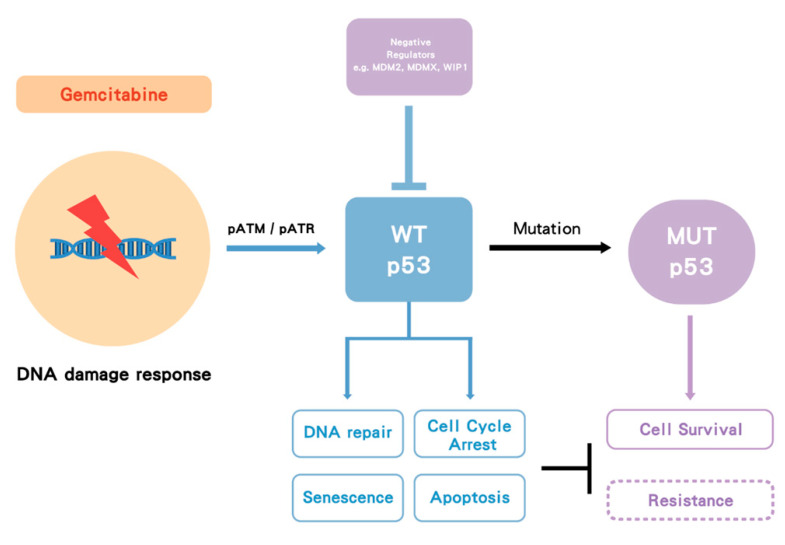
**The proposed model of p53 impact on gemcitabine treatment.** Gemcitabine induces a DNA damage response followed by activation of pATM/pATR and p53. Wild-type (WT) p53 activation can facilitate DNA repair, cell cycle arrest, apoptosis and senescence. Once p53 becomes mutated (MUT), it loses its WT function, and some mutant forms may gain function, which increases cell survival. In some cases, WT p53 is suppressed by its negative regulators such as MDM2/MDMX. Either p53 mutation or suppression by MDM2 may increase gemcitabine resistance. Dashed box indicates hypothesis.

**Figure 6 biomolecules-10-01474-f006:**
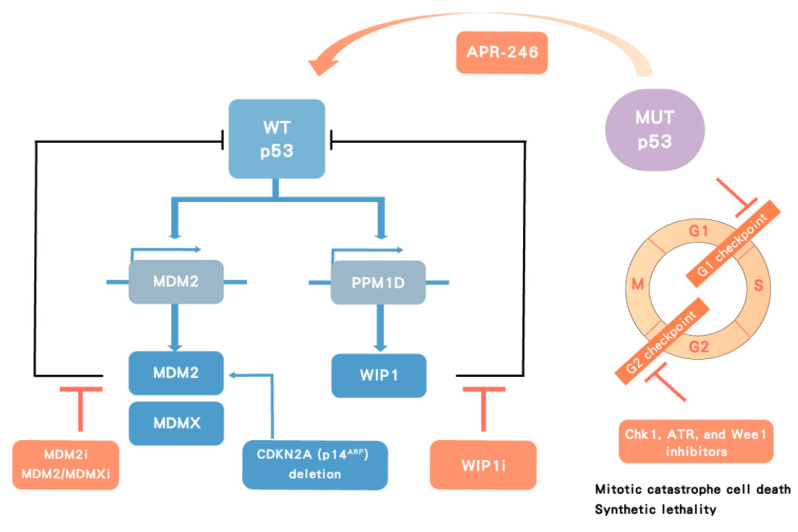
**Therapeutic strategy targeting p53.** In p53 wild-type (WT) cancers, p53 transactivates its negative regulators, MDM2 and wild-type p53-induced phosphatase 1 (WIP1), which in turn inhibit p53 function. MDMX cooperates with MDM2 to degrade p53. MDM2 inhibitors (MDM2i), dual MDM2/MDMX inhibitors (MDM2/MDMXi) and WIP1 inhibitors (WIP1i) target negative regulators of p53 leading to p53 stabilization and activation. In contrast, p53 mutant (MUT) tumors can potentially be treated with a direct activator, such as APR-246, or indirectly with Chk1, ataxia telangiectasia related (ATR), and Wee1 inhibitors, which lead to cell death by mitotic catastrophe in MUT p53 cancers.

**Table 1 biomolecules-10-01474-t001:** Selected clinical trials targeting the p53 pathway in solid cancers.

Class	Drugs	Phase	Tumor Types	ClinicalTrials.gov
p53 activator	APR-246 + Pembrolizumab	1/2	Solid Cancers	NCT04383938
Wee1 inhibitor	AZD1775	1	Solid Cancers	NCT01748825 NCT04462952 NCT02482311
ATR inhibitor	AZD6738 + Durvalumab	2	Bile Duct Cancer	NCT04298008
MDM2 inhibitor	HDM201	1	WT-p53 Solid Cancers	NCT02143635
	Idasanutlin	1	Solid Cancers	NCT03362723
MDM2/MDMX Inhibitor	ALRN-6924 + Paclitaxel	1	WT-p53 Solid Cancers	NCT03725436
	ALRN-6924	1	WT-p53 Pediatric Cancer	NCT03654716

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
