# Peer review of "Targeting P53 as a Future Strategy to Overcome Gemcitabine Resistance in Biliary Tract Cancers"

_biomolecules, 2020, doi:10.3390/biom10111474_

Round 1
Reviewer 1 Report
The manuscript “Targeting p53 as a future strategy to overcome gemcitabine resistance in biliary tract cancers” by Wu et al. summarizes the current knowledge about p53-mediated responses to gemcitabine chemotherapies and their abrogation by p53 mutations in BTCs, as well as therapeutic options to target p53.
After a brief overview of current treatment options in BTCs, and a concise summary of p53 function, the authors review the role of p53 in BTCs, with data obtained from the cbioportal online database. Next, the authors summarize what is known about p53´s role in resistance to gemcitabine, and conclude with therapeutic strategies to target (mutant) p53.
- The major weakness of the manuscript is that a specific focus on the role of p53 on gemcitabine resistance in BTCs and its underlying mechanisms is not a major part of the manuscript due to the paucity of relevant studies, as the authors themselves point out. For example, the section on gemcitabine resistance largely focuses on studies on pancreatic cancers. Likewise, the strategies to reactivate mutant p53 which are reviewed in the manuscript apply to other forms of cancers as well and are not specific for BTCs. However, I am aware that due to the aforementioned reason this concern probably cannot be addressed adequately.
- Apart from that, there are some additional points that need to be addressed before being acceptable for publication :
Line 87 : “MDM2, a p53-specific E3 ubiquitin ligase”. Actually, MDM2 is not “p53-specific”, since it has other targets/functions apart from regulation of p53, even though p53 regulation may be regarded as a very critical one.
Line 97 : There are several reports that DDR-induced phosphorylation of MDM2 as well abrogates MDM2-mediated regulation of p53.
Line 120 : Please include some information about p53 mutations frequently targeting the DNA binding domain of p53 (either here or in section 2), with a reference to classical “hotspot mutations”, and please highlight some of these in Fig. 2D.
Line 159 : Reference is missing
Line 184 : Regarding CD44 and MDM2 : In Ref 61, co-expression of CD44 and MDM2 predicts particularly poor prognosis. However, their co-expression actually is contradictory to the authors´ assumption to see them as surrogate markers for p53 status.
MDM2 is known to be regulated by other factors such as FLI1 and MYCN, and depicting p53 as the sole regulator of MDM2 may be too simplistic. As the authors themselves point out, the “real relationship between CD44 / MDM2 / TP53” in these studies can only be speculated about.
Line 228 : “MiroRNA(miR) profiling in chemoresistant pancreatic cancer suggested involvement in pathways controlling cell cycle progression and cell death. This was associated with significant MRP-1 and Bcl-2 overexpression in resistant cell clones and of mutant p53 in one clone”
The relationship between p53, microRNAs, their target mRNAs and chemoresistance should be explained more clearly.
Line 273 : Could the authors provide a reference for the study ? Is it NCT03588078 ?
Line 359 : Reference is missing
- Figure 5 : For clarity, please change to “Negative regulators e.g. MDM2 ….” in the figure.
- Please check proper use of human gene nomenclature (where applicable) throughout the manuscript (incl. figures)
- There should be explicit references in the text to the clinical trials table apart from the one in line 371, for example in line 346.
Author Response
Comments and Suggestions for Authors
The manuscript “Targeting p53 as a future strategy to overcome gemcitabine resistance in biliary tract cancers” by Wu et al. summarizes the current knowledge about p53-mediated responses to gemcitabine chemotherapies and their abrogation by p53 mutations in BTCs, as well as therapeutic options to target p53.
After a brief overview of current treatment options in BTCs, and a concise summary of p53 function, the authors review the role of p53 in BTCs, with data obtained from the cbioportal online database. Next, the authors summarize what is known about p53´s role in resistance to gemcitabine, and conclude with therapeutic strategies to target (mutant) p53.
- The major weakness of the manuscript is that a specific focus on the role of p53 on gemcitabine resistance in BTCs and its underlying mechanisms is not a major part of the manuscript due to the paucity of relevant studies, as the authors themselves point out. For example, the section on gemcitabine resistance largely focuses on studies on pancreatic cancers. Likewise, the strategies to reactivate mutant p53 which are reviewed in the manuscript apply to other forms of cancers as well and are not specific for BTCs. However, I am aware that due to the aforementioned reason this concern probably cannot be addressed adequately.
Reply: Thank you for your comments and understanding the limited availability of specific BTC studies. It is because studies on BTCs are limited and effective treatment is lacking that this review is aimed to provide possible options for future treatment targeting the p53 pathway in BTCs.
- Apart from that, there are some additional points that need to be addressed before being acceptable for publication :
Line 87 : “MDM2, a p53-specific E3 ubiquitin ligase”. Actually, MDM2 is not “p53-specific”, since it has other targets/functions apart from regulation of p53, even though p53 regulation may be regarded as a very critical one.
Reply: We agree with reviewer’s comments that MDM2 is critical for p53 but not p53-specific so we removed “p53-specific” in the revised manuscript. However, it is clear from MDM2 gene knockout studies in which it is developmentally lethal, yet can be rescued by co-deletion of Tp53, that p53 is the major target. Furthermore, in cells that are devoid of p53 there is little or no detectable expression of MDM2 and in such circumstances, in the absence of p53, MDM2 inhibitors have no phenotypic consequences.
Line 97 : There are several reports that DDR-induced phosphorylation of MDM2 as well abrogates MDM2-mediated regulation of p53.
Reply: Thank you for your comments. We agree with this and have included cited mention of phosphorylation of MDM2 in the revised manuscript.
Line 120 : Please include some information about p53 mutations frequently targeting the DNA binding domain of p53 (either here or in section 2), with a reference to classical “hotspot mutations”, and please highlight some of these in Fig. 2D.
Reply: This is a good suggestion and we have included this important information in the revised manuscript and highlight it in the legend of figure 2D.
Line 159 : Reference is missing
Reply: We have added the reference in the revised manuscript.
Line 184 : Regarding CD44 and MDM2 : In Ref 61, co-expression of CD44 and MDM2 predicts particularly poor prognosis. However, their co-expression actually is contradictory to the authors´ assumption to see them as surrogate markers for p53 status.
MDM2 is known to be regulated by other factors such as FLI1 and MYCN, and depicting p53 as the sole regulator of MDM2 may be too simplistic. As the authors themselves point out, the “real relationship between CD44 / MDM2 / TP53” in these studies can only be speculated about.
Reply: Thank you for your comments. CD44 was found to be associated with p53 mutation but not a surrogate marker for p53 mutation. Therefore, wild type p53 tumors may still express CD44. Although FLI1, MYCN, and other transcript factors (TFs) were reported to transactivate MDM2, p53 is still the most critical regulator for MDM2. This has been validated in numerous studies and represents hard data. To avoid misunderstanding, we have revised this section and emphasized that the p53 status in CD44/MDM2 co-expressed tumors is unknown.
Line 228 : “MiroRNA(miR) profiling in chemoresistant pancreatic cancer suggested involvement in pathways controlling cell cycle progression and cell death. This was associated with significant MRP-1 and Bcl-2 overexpression in resistant cell clones and of mutant p53 in one clone”
The relationship between p53, microRNAs, their target mRNAs and chemoresistance should be explained more clearly.
Reply: Thank you for your comments. We have rewritten this section to clarify the relationship between p53, microRNAs and proteins in the revised manuscript.
Line 273 : Could the authors provide a reference for the study ? Is it NCT03588078 ?
Reply: Yes, It is the trial, NCT03588078. Only a meeting abstract is available so we added this reference in the revised manuscript.
Line 359 : Reference is missing
Reply: For the development of GSK2830371 and WIP1i, no reference is available. This information is from private communication with staffs in the GSK Company. One possible reason is the unsuitability of GSK280371 for clinical use from the Pechackova review. (J Mol Med (2017) 95:589–599, DOI 10.1007/s00109-017-1536-2). We have cited this reference [102] in the revised manuscript.
- Figure 5 : For clarity, please change to “Negative regulators e.g. MDM2 ….” in the figure.
Reply: We have changed this in the revised manuscript.
- Please check proper use of human gene nomenclature (where applicable) throughout the manuscript (incl. figures)
Reply: We have checked and revised those in the revised manuscript.
- There should be explicit references in the text to the clinical trials table apart from the one in line 371, for example in line 346.
Reply: We have added the references in the revised manuscript.
Reviewer 2 Report
In this review the authors summarized the status of oncosuppressor p53 in biliary tract cancers (BTCs) in order to highlight how to overcome gemcitabine resistance. To this aim they summarize how to target mutant p53 or wild-type p53 inhibited by other molecular mechanisms.
The review is well balanced and the references are up-to-date.
Only few points should be addressed, in particular regarding the description of the figures.
The 4 panels in figure 2 need to be described in the text.
Figurere 3B should be better described in the text.
What is APR-246? It should be briefly described and a reference added.
The Table should have a number (i.e., Table 1).
Author Response
Reviewer 2
Comments and Suggestions for Authors
In this review the authors summarized the status of oncosuppressor p53 in biliary tract cancers (BTCs) in order to highlight how to overcome gemcitabine resistance. To this aim they summarize how to target mutant p53 or wild-type p53 inhibited by other molecular mechanisms.
The review is well balanced and the references are up-to-date.
Only few points should be addressed, in particular regarding the description of the figures.
The 4 panels in figure 2 need to be described in the text.
Reply: We have added the description and cited corresponding panels in the text of the revised manuscript.
Figurere 3B should be better described in the text.
Reply: We have added the description and cited corresponding panels in the text of the revised manuscript.
What is APR-246? It should be briefly described and a reference added.
Reply: We have added the description and cited corresponding reference [70] in the “5.1.1. Direct targeting of mutant p53” of revised manuscript.
The Table should have a number (i.e., Table 1).
Reply: There was only one table in this review so we did not number this. However, we have now numbered this table in the revised manuscript.
Round 2
Reviewer 1 Report
All points raised by this reviewer have been addressed in the revised version.
However, please re-check lines 241ff :
“were predicted targets from dysregulated miRs” ?
“was observed in both resistant cell clones” ?
The formats of human gene/RNA/protein symbols are not uniform (e.g. WEE1, Wee1); however, I realized that this is also not the case in many of the respective references.
Author Response
Comments and Suggestions for Authors All points raised by this reviewer have been addressed in the revised version. However, please re-check lines 241ff : “were predicted targets from dysregulated miRs” ? “was observed in both resistant cell clones” ? Reply: We have corrected these in the revised manuscript. In addition, we revised this paragraph to make this paragraph easier to read. The formats of human gene/RNA/protein symbols are not uniform (e.g. WEE1, Wee1); however, I realized that this is also not the case in many of the respective references. Reply: We have corrected these including Wee1, Chk1 as uniform in the revised manuscript. In addition, we used TP53, PPM1D, MDM2, CDKN1A indicate genes and p53, WIP1, MDM2 indicate protein in the whole manuscript. However, we did not revise the format used in the list of published references. This would not be appropriate and incorrect.